# The Observation of Cellular Precipitation in an Ni_36_Co_18_Cr_20_Fe_19_Al_7_ High-Entropy Alloy after Quenching and Annealing

**DOI:** 10.3390/ma15196613

**Published:** 2022-09-23

**Authors:** Gurumayum Robert Kenedy, Korir Rosemary Chemeli, Wei-Chun Cheng

**Affiliations:** Department of Mechanical Engineering, National Taiwan University of Science and Technology, 43 Keelung Road, Section 4, Taipei 106, Taiwan

**Keywords:** high-entropy alloy, L1_2_ phase, cellular precipitation

## Abstract

High-entropy alloys (HEAs) comprise a minimum of five major elements. These alloys show some special characteristics, such as excellent mechanical and high temperature properties. The development of the HEAs requires a knowledge of phase transformations during alloy making procedures. The phase transformations of an Ni_36_Co_18_Cr_20_Fe_19_Al_7_ HEA were studied in this research. The alloy underwent hot forging, cold rolling, annealing at and quenching from 1323 K, and isothermal holding at 873 K. The alloy is a single face-centered cubic (FCC) phase in the as-quenched condition. After annealing at 873 K, not only fine coherent L1_2_ particles precipitated homogeneously in the FCC matrix, but lamellae of FCC and L1_2_ phases also developed from the grain boundaries. Both lamellar FCC and L1_2_ grains have a cubic-on-cubic orientation relationship (OR). The composition of the lamellar L1_2_ phase is Ni_60_Co_8_Cr_6_Fe_6_Al_20_, and that of the lamellar FCC phase is Ni_31_Co_15_Cr_28_Fe_21_Al_4_. Cellular precipitation occurs in the HEA, and the high-temperature FCC (γ) transforms to a lamella of low-temperature FCC (γ_1_), and an L1_2_ phase, i.e., γ → γ_1_+L1_2_.

## 1. Introduction

High-entropy alloys are composed of a minimum of five components, whose concentrations are approximately between 5 and 35 at.%. HEAs show some distinct features, for example, excellent mechanical strength, corrosion resistance, wear resistance, and high temperature oxidation properties. Since the high-entropy alloys may have various outstanding properties, they are of considerable importance in engineering applications [1,2,3]. In the study of the HEAs containing aluminum by transmission electron microscope (TEM) observations, phase transformations—such as precipitation transformation [4,5,6], spinodal decomposition [7,8,9,10,11], and ordering reaction [12,13]—were found, and their product phases were L1_2_ [14,15,16], and B2 [17,18,19,20,21,22] superlattices.

The precipitation transformation begins with the appearance of a second phase (β), from the supersaturated matrix phase (α) of an alloy after cooling from high temperature. The high-temperature phase has, thus, been changed to the low-temperature phase (α_1_). The phase transformation is as follows: α → α_1_ + β. The concentration of the low-temperature α_1_ phase is different from the high-temperature α phase, but with the same crystal structure and orientation. The second phase (β) prefers to precipitate at the grain boundary and is called grain-boundary precipitate, and it has a different composition and a different crystal structure from the matrix phase. A necessary condition for the occurrence of the precipitation transformation is the long-distant diffusion of atoms through the matrix phase. Grain-boundary precipitation always results in the appearance of separate grains, allotriomorphs, Widmanstätten plates, and/or needles [23].

A special type of precipitation transformation is known as cellular precipitation. There are three distinct characteristics for the occurrence of cellular precipitation, which are as follows: The first characteristic is that the product phases are in the form of lamellar grains, which develop from the grain boundary. The second feature is that the lamellar α_1_ grains have the same orientation as that of one neighboring α grain, but a different orientation from the other. The third distinctive characteristic is that the grain boundary migrates, i.e., the grain boundary moves with the growing fronts of the cellular product phases; therefore, the newly formed lamellae nucleate at the grain boundary, where the α1 grain has the same orientation as one neighboring α grain and grows, simultaneously, into the other neighboring α grain with the movement of the original grain boundary [23,24,25]. Cellular precipitation occurs in some alloy steels. For example, in Fe-Mn-Al steel, after quenching and annealing, cellular precipitation occurs, and high-temperature FCC decomposes into lamellae of low-temperature FCC and κ-carbide, at the grain boundary [26]. The motivation for this paper was to report the occurrence of interesting cellular precipitation in an Al-containing HEA.

## 2. Experimental Procedures

By induction melting in a vacuum chamber, a 10-kg Ni_36_Co_18_Cr_20_Fe_19_Al_7_ ingot was formed by melting the following metals with high purity together: nickel, cobalt, chromium, iron, and aluminum. To ensure the uniform alloy composition, the ingot was homogenized at 1473 K, for 4 h, under an argon-protected atmosphere; hot forged and re-annealed at 1473 K, for at least 3 cycles; cut into plates; annealed at 1473 K for 4 h; and quenched. The plates were cold rolled into 2 mm plates at room temperature, and cut into samples, measuring 10 mm × 10 mm. In an argon atmosphere, samples were heated at 1323 K, for 1 h, and then water quenched. Under vacuum, specimens were sealed in quartz tubes and held isothermally at 873 K, for 24 h. After metallographic sample preparation, the samples were observed by using an optical microscope and a Jeol JEM 6500F high-resolution, field-emission scanning electron microscope (SEM, Jeol Ltd., Akishima, Tokyo, Japan). The crystal structure of the as-quenched HEA in powder form, sealed in a vacuum quartz tube, was analyzed by adopting X-ray diffraction (XRD) in the Taiwan photon synchrotron light source, of national synchrotron radiation center, Hsinchu, Taiwan, operated at 15 keV and a wavelength of 0.082656 nm. The sample, in the form of alloy plates annealed at 873 K, were also examined by XRD, in a Rigaku DMAX-B X-ray diffractometer, operated at a power of 12 kW. TEM samples were prepared by thinning 80 μm thin foils, punching them into discs with a diameter of 3 mm, and electro-polishing them in a 90% acetic acid and 10% perchloric acid solution. The TEM, with the brand of FEI (Thermo Fisher Scientific) Talos F200XG2 (Thermo Fisher Scientific, Waltham, MA, US), equipped with the energy dispersive spectroscopy (EDS) was utilized to analyze the TEM samples. The TEM operation voltage was 200 kV.

## 3. Results and Discussion

Figure 1 shows the results of the investigation on the Ni_36_Co_18_Cr_20_Fe_19_Al_7_ HEA, after the solution treatment at 1323 K. An optical micrograph (OM) and a secondary electron image (SEI) from the SEM in Figure 1a,b, respectively, illustrate that the HEA has similar grains with annealing twins. Only FCC peaks were detected by the synchrotron-based XRD, as shown in Figure 1c. Therefore, the HEA is a single phase of FCC after the solution treatment at 1323 K, and the lattice parameter of the FCC phase is approximately 0.3589 nm ± 5 nm.

After understanding the constituent phase of the as-quenched HEA, we examined the crystal structures of the HEA that was held isothermally, at 873 K. The results from the OM, SEI, and XRD studies are shown in Figure 2a–c, respectively. From the OM observation in Figure 2a, the FCC grains with twins remain as the matrix phase. However, besides the matrix FCC phase, the other phases precipitate at the grain boundaries. Separate grains and lamellar colonies appear in the alloy simultaneously, as shown in Figure 2b. The analysis of the XRD, as shown in Figure 2c, reveals that the HEA has a major phase of FCC, and that the crystal structures of the second phases, precipitating at the grain boundaries, cannot be clearly identified. Therefore, we applied the TEM to study the grain boundary precipitates in detail.

The TEM analysis of the HEA annealed at 873 K is illustrated in Figure 3. We focused the TEM study on the grain-boundary colony, as well as the FCC matrix. A selected area diffraction pattern (SADP), taken on the FCC matrix, is shown in Figure 3a. We applied the smallest SADP aperture, with a diameter of approximately 100 nm, to take the SADP. The zone axis of the FCC phase on the SADP in Figure 3a is along the [011] direction. The SADP in Figure 3a reveals the appearance of extra superlattice (100) reflections. These reflections are from the L1_2_ superlattice phase. The zone axis of the L1_2_ phase on the SADP is also along the [011] direction. Therefore, there is a cubic-on-cubic orientation relationship (OR) between the L1_2_ precipitate and the FCC matrix, i.e., [011]_L12_//[011]_γ_ and (200)_L12_//(200)_γ_. As shown in Figure 3a, the Miller indexes of the L1_2_ reflections are underlined to differentiate them from those of the FCC reflections. In Figure 3b, a dark-field (DF) image, taken from the superlattice reflection of the L1_2_ (100), manifests fine coherent particles that are distributed uniformly in the FCC matrix. Thus, the L1_2_ superlattice phase precipitates homogeneously in the FCC matrix, as fine coherent particles during annealing. The corresponding bright-field (BF) image in Figure 3c shows that the fine L1_2_ particles associated with the strain energy field can also be clearly seen. We estimated the particle shape as a sphere, and the spherical particles have a diameter of approximately 2 nm. The L1_2_ phase has a Cu_3_Au crystal structure, which is derived from its parent FCC phase, and has the space group, Pm$\bar{3}$m. In the Al-content HEAs, fine coherent L1_2_ particles precipitating homogeneously in the FCC phase were reported previously [14,15,16].

In addition to the homogeneous precipitation of the fine L1_2_ particles in the FCC grains, grain-boundary precipitates not only appear in the form of separate grains, but also as colonies from the SEI observation, as shown in Figure 2b. The TEM investigation of the colony is shown in Figure 3d–f. An SADP (taken on the lamellae) similar to that in Figure 3a is shown in Figure 3d. The zone axes of both the FCC and L1_2_ superlattice on the SADP are all along the [011] directions. A DF image in Figure 3e, taken from the 100 superlattice reflection of the L1_2_ phase, illustrates the locations of the lamellar L1_2_ grains (with a white contrast) in the lamellae. It is worth noting that, in the colony, the dark lamellar grains between the white ones are FCC. The corresponding BF image in Figure 3f presents the following three sections: an FCC grain at the left-hand side, with a bright contrast; a colony in the middle, with a dark contrast; and an FCC grain at the right-hand side, with the same contrast as the colony. The colony is located between two FCC grains. During the TEM operation at the same tilting condition, the same contrast for both the lamellae and FCC grain demonstrated that both orientations are identical, as was also confirmed by both SADPs. This means that the lamellar FCC grains nucleate at the grain boundary, with the same orientation as the FCC grain at the right-hand side and have a cubic-on-cubic OR, along with the lamellar L1_2_ grains. Thus, the fact that colonies of lamellar FCC and L1_2_ grains develop from the grain boundaries has been confirmed. The precipitation of the lamellar FCC and L1_2_ phases is achieved via cellular precipitation [23,24,25]. Therefore, cellular precipitation occurs in the HEA, and the phase transformation is as follows: γ → γ_1_ + L1_2_. The lamellar L1_2_ and γ_1_ FCC grains are the product phases of the cellular precipitation.

The lamellae of both the FCC and L1_2_ phases nucleate and grow simultaneously at the grain boundary. The lamellar FCC grains have the same orientation as that of the right FCC matrix and have a cubic-on-cubic OR with the lamellar L1_2_ grains. The lamellar phases grow side-by-side, forward, and to the left of the FCC grain. Figure 3f illustrates more evidence for the occurrence of cellular precipitation. The BF image in Figure 3f shows the original grain boundary between two FCC grains, located near the centered vertical line, or, in other words, located at the right-hand side boundary of the colony. The lamellae nucleate at the grain boundary and grow into the FCC grain at the left-hand side, with a curved boundary. As shown in Figure 3e, it is not only fine white L1_2_ particles—which are the same as those in Figure 3b—that appear uniformly in the right FCC grain, but white L1_2_ lamellar grains also appear in the lamellae. This reveals more evidence that the lamellar FCC grains in the newly formed lamellae have the same orientation as that of the right-neighboring FCC grain. However, the lamellae grow into the left-neighboring FCC grain in a different orientation. Therefore, all the above satisfy the distinct characteristics of cellular precipitation, i.e., the lamellar FCC grains, along with the L1_2_ grains, nucleate at the grain boundary with the same orientation as one neighboring FCC grain and grow into the other neighboring FCC grain (in a different orientation), with the movement of the grain boundary. These are the distinctive features of cellular precipitation [23]. Therefore, the occurrence of cellular precipitation for the transformation of the high-temperature FCC phase to the low-temperature FCC and L1_2_ phases has been addressed for the first time in the Al-contained NiCoCrFe HEAs.

The concentrations of the lamellar phases in the colonies were studied by EDS, in the STEM mode of the Tolas TEM. For example, the composition line scans of the lamellar grains are shown in Figure 4. Figure 4a illustrates a BF image covering a section of the lamellae, in which the thinner lamellar grains with a dark contrast are the L1_2_ phase and the thicker lamellar grains with a grey contrast are the FCC phase. The concentrations of the constituent elements were measured by the line scans on the white solid line, as drawn on the BF image in Figure 4a. The concentration curves for the following elements: Ni, Al, Co, Cr, and Fe, are shown in Figure 4b–f, respectively. The average composition (at.%) of the lamellar L1_2_ phase in the following sequential elements of the Al, Co, Cr, Fe, and Ni elements is approximately 19.9, 8.1, 5.8, 6.4, and 59.8 at.%, respectively, and the FCC is approximately 4.3, 15.3, 28.4, 21.2, and 30.8 at.%, respectively. The margin of error in the elemental composition is approximately 10%. In the other form, the lamellar L1_2_ phase has the following composition: Ni_60_Co_8_Cr_6_Fe_6_Al_20_, and the FCC grain is as follows: Ni_31_Co_15_Cr_28_Fe_21_Al_4_. This result shows that the lamellar L1_2_ grains are rich in Ni and Al, and lean in Co, Cr, and Fe, and that the lamellar FCC grains are vice versa. The ratio of the concentration of Ni to that of Al in the lamellar L1_2_ phase is approximately three, and it means that the L1_2_ phase is approaching the Ni_3_Al phase.

## 4. Conclusions

The phase transformations of the Ni_36_Co_18_Cr_20_Fe_19_Al_7_ HEA have been studied in this research paper. The alloy underwent hot forging, cold rolling, annealing at and quenching from 1323 K, and isothermal holding, at 873 K. The as-quenched HEA is a single FCC. After the 873-K isothermal holding, the coherent fine L1_2_ particles precipitated homogeneously in the FCC matrix. In addition, cellular precipitation occurred, resulting in the transformation of the high-temperature FCC phase into lamellae of the low-temperature FCC phase and L1_2_ phase. The colonies nucleated and grew from the grain boundaries. The grain boundary moved with the growing tips of the newly formed lamellar FCC and L1_2_ phases, and the lamellar FCC grains had the same orientation as that of one neighboring FCC grain and grew into the other neighboring FCC grain, with a different orientation. The lamellar L1_2_ grains had the following composition: Ni_60_Co_8_Cr_6_Fe_6_Al_20_, and the FCC grains had the following composition: Ni_31_Co_15_Cr_28_Fe_21_Al_4_. The lamellar L1_2_ phase was rich in Ni and Al, and lean in Co, Cr, and Fe, and the lamellar FCC phase was contrary to this.

## Figures and Tables

**Figure 1 materials-15-06613-f001:**
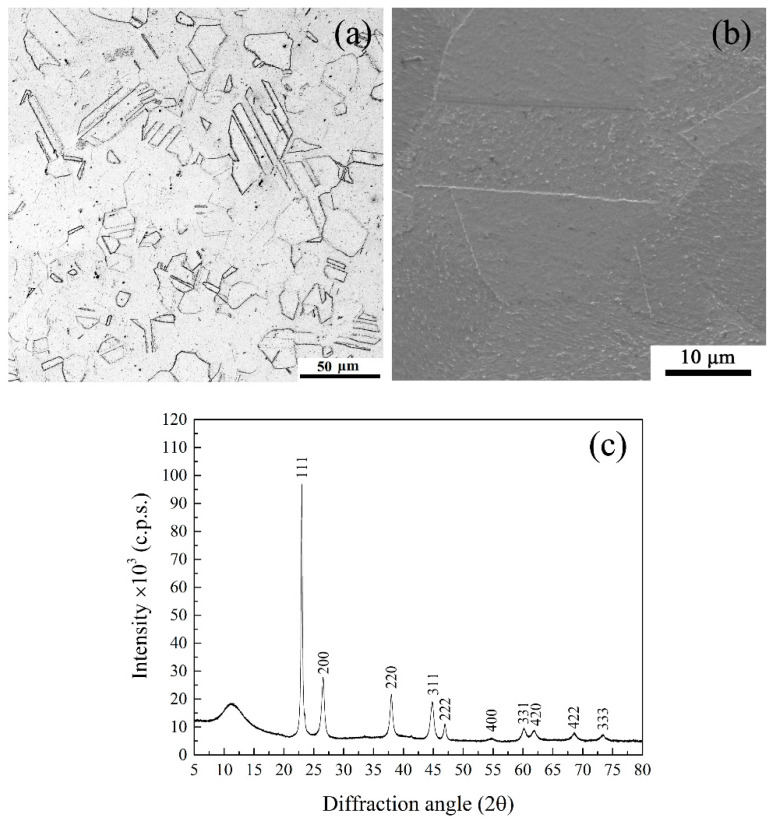
The investigation of the HEA after solution treatment at 1323 K, for 1 h: (**a**) OM, (**b**) SEI, and (**c**) XRD. Note that the unmarked broad peak at approximately 11 degrees of the diffraction angle is the signal from the quartz tube.

**Figure 2 materials-15-06613-f002:**
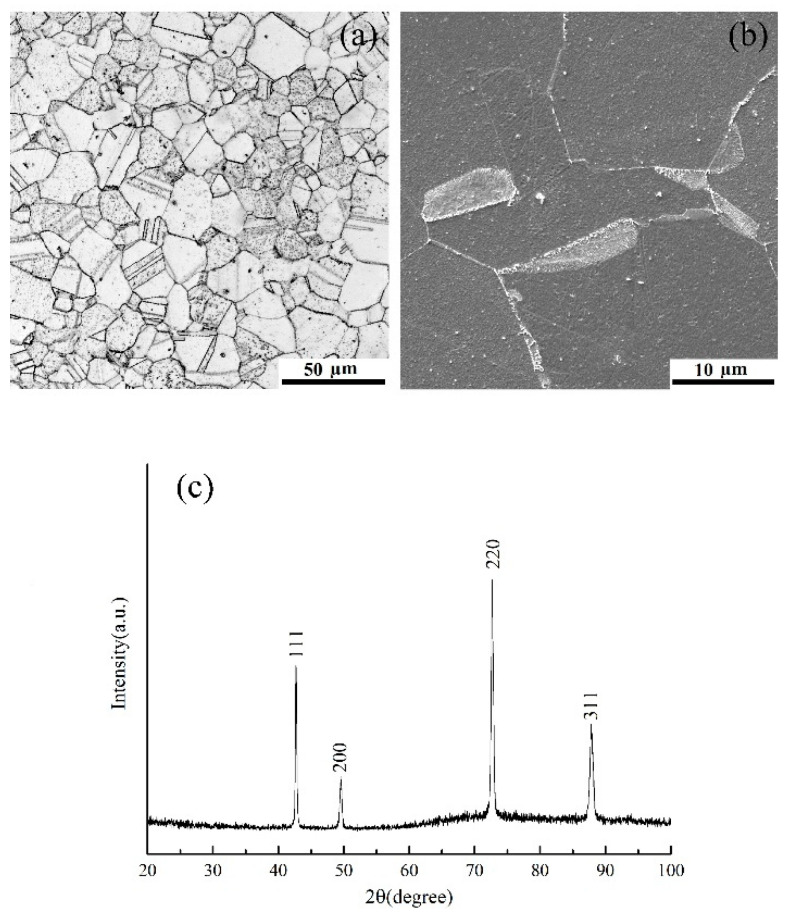
(**a**) OM, (**b**) SEI, and (**c**) XRD of the HEA after quenching, from 1323 K, and isothermal holding at 873 K, for 24 h.

**Figure 3 materials-15-06613-f003:**
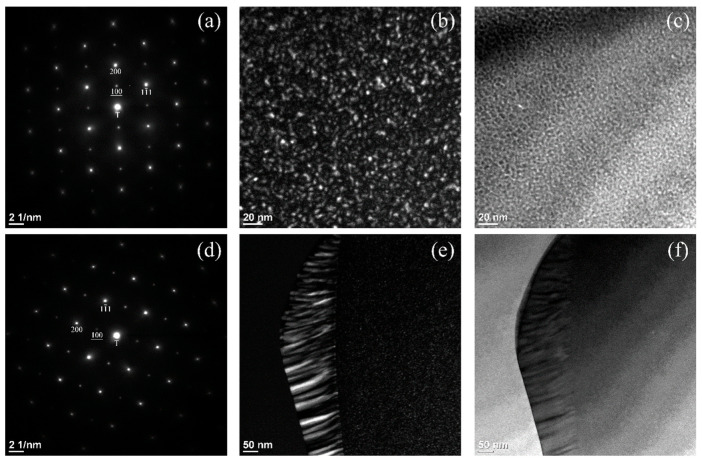
The TEM analysis of the HEA with the same heat treatment as Figure 2: (**a**) [011] SADP taken from the FCC matrix; (**b**) DF image taken from the L1_2_ (100) reflection in (**a**); (**c**) the BF image corresponding to the DF image in (**b**); (**d**) [011] SADP taken from the lamellae at the grain boundary; (**e**) the DF image taken from the L1_2_ (100) reflection; and (**f**) the BF image corresponding to (**e**).

**Figure 4 materials-15-06613-f004:**
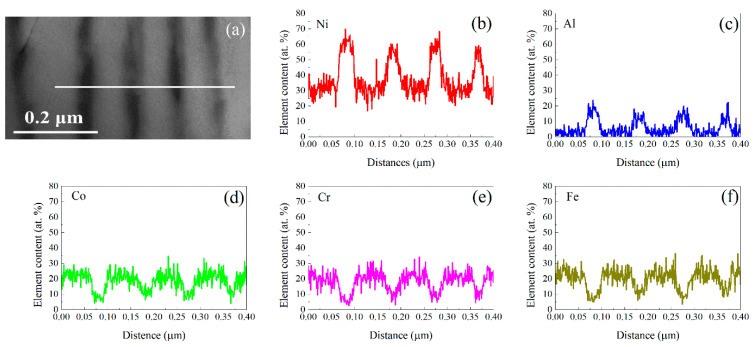
The TEM study of the HEA in STEM mode to illustrate the composition differences between the lamellar grains: (**a**) BF image taken on the lamellae. The concentration line scans by EDS, along the white solid line marked in (**a**), as follows: (**b**) Ni, (**c**) Al, (**d**) Co, (**e**) Cr, and (**f**) Fe. The specimen had the same heat treatment as that in Figure 2.

## Data Availability

Not applicable.

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
