# Peer review of "The Observation of Cellular Precipitation in an Ni36Co18Cr20Fe19Al7 High-Entropy Alloy after Quenching and Annealing"

_materials, 2022, doi:10.3390/ma15196613_

Round 1

Reviewer 1 Report

The study was carried out competently, but it presents the known facts of the formation of the microstructure of high-intropy alloys.

1. The excursion into the literature, in my opinion, should be improved so that there is no dissonance when comparing the information given in the sections "Introduction", "Discussion" and "Conclusions".

2. The conclusion of the authors that cellular separation leads to the transformation of high-temperature austenite into plates of low-temperature austenite causes doubt. The grain boundary does move, but in my opinion these are banal twins of deformation, which appear just in the places where secondary phases are separated. I would like to hear a reasoned response from the authors on this issue. With the formation of deformation twins along the boundaries of austenite grains, a change in the orientation of the crystal lattice will necessarily occur.

Author Response

The study was carried out competently, but it presents the known facts of the formation of the microstructure of high-entropy alloys.

  1. The excursion into the literature, in my opinion, should be improved so that there is no dissonance when comparing the information given in the sections "Introduction", "Discussion" and "Conclusions".

Reply: We appreciate the reviewer kindly remind us to improve the statement of the text. Therefore, we have gone through all the text of the paper to smoothen the information we mentioned several times.

  1. The conclusion of the authors that cellular separation leads to the transformation of high-temperature austenite into plates of low-temperature austenite causes doubt. The grain boundary does move, but in my opinion these are banal twins of deformation, which appear just in the places where secondary phases are separated. I would like to hear a reasoned response from the authors on this issue. With the formation of deformation twins along the boundaries of austenite grains, a change in the orientation of the crystal lattice will necessarily occur.

Reply: Yes. It is possible for the FCC twins appear in the lamellae of FCC and L12 phases. We have found this phenomenon in other alloy systems. However, at this moment, we did not discover the twins appear in the lamellae in Fig. 3 from the observation of the SADPs. It might appear from another <110> orientations. We observed only the lamellar FCC grains are with the same orientation as that of the FCC grain at the right-hand side as shown in Fig. 3(f).

Reviewer 2 Report

Mayor issues:

    The manuscript persents two samples: an as-quenched and a heat treated one. However, several parts of hte paper suggest that there are more samples investigated (eg. lines 16, 93: "temperatures"; line 93: "for example"). Please correct the article to be clear that is about two samples.
    Since the article is not about iron/steel and there is no FCC-BCT (austenite-like martensite-like) transition, the word "austenite" should be avoided refering to the FCC phase.
    Lines 98-101 and figure 2: XRD results should be presented.
    Lines 174-191: how were the atomic ratios measured (eg. EDS, EELS)? Please include it to the "Experimental procedures" part.

Minor issues:

    Line 70: value of the temperature should be mentioned (873 K).
    Line 75: the wavelength is only 0.082656 nm

Minor questions:

    Line 66: "several cycles": how many?
    Line 87: what is the margin of error?
    Line 109, figure 3: how large was the are where SADP was taken?
    Lines 183-184: what is the margin of error?

Typos, format remarks:

    Line 38: ":" instead of "."
    Line 45: "Widmanstätten" with only two "t"
    Line 73: "X-ray", capitalized.
    Line 74: "keV", lower case
    Figure 1: typo at the end of the caption.
    Line 172: NiCoCrFe
    Lines 174-175: simply "in STEM mode" or "by STEM".
    Figure 4: subfigure keys (Ni, Al, Co...) should be placed in fix position in the top left corners.

Author Response

The manuscript presents two samples: an as-quenched and a heat treated one. However, several parts of the paper suggest that there are more samples investigated (eg. lines 16, 93: "temperatures"; line 93: "for example"). Please correct the article to be clear that is about two samples.

Reply: Thanks for the reviewer pointing out the mistakes. We fixed them with only two samples included in this article. The places with the corrections are marked with a red color.

Since the article is not about iron/steel and there is no FCC-BCT (austenite-like martensite-like) transition, the word "austenite" should be avoided referring to the FCC phase.

Reply: We replaced all the austenite into FCC or FCC phase (matrix).

 Lines 98-101 and figure 2: XRD results should be presented.

Reply: We added the XRD as Fig. 2(c). However, it was not from not from the Synchrotron radiation, but the XRD in a Rigaku DMAX-B x-ray diffractometer operated at a power of 12 kW.

Lines 174-191: how were the atomic ratios measured (eg. EDS, EELS)? Please include it to the "Experimental procedures" part.

Reply: They were measured by the EDS which was added and discussed in the Experimental procedures: (The TEM with the brand of FEI (Thermo Fisher Scientific) Talos F200XG2 equipped with the energy dispersive spectroscopy (EDS) was utilized to analyze the TEM samples.)

Minor issues:

Line 70: value of the temperature should be mentioned (873 K).

Reply: We fixed it.

Line 75: the wavelength is only 0.082656 nm

Reply: It is amazing that the reviewer noticed the error. We appreciated the careful examination of the mistakes by the reviewer. We fixed the typing mistake.

Minor questions:

Line 66: "several cycles": how many?

Reply: at 1473 K for at least 3 cycles.

Line 87: what is the margin of error?

Reply: 5 nm.

Line 109, figure 3: how large was the area where SADP was taken?

Reply: The smallest SADP aperture with a diameter 100 nm: We applied the smallest SADP aperture with a diameter about 100 nm to take the SADP.

Lines 183-184: what is the margin of error?

Reply: The margin of error in the elemental composition is about 10 %.

Typos, format remarks:

Line 38: ":" instead of "."

Reply: We fixed it: as follows: a à a1 + b.

Line 45: "Widmanstätten" with only two "t"

Reply: We fixed it.

Line 73: "X-ray", capitalized.

Reply: We fixed it.

Line 74: "keV", lower case

Reply: We fixed it: keV.

Figure 1: typo at the end of the caption.

Reply: We fixed it.

Line 172: NiCoCrFe

Reply: We fixed it.

Lines 174-175: simply "in STEM mode" or "by STEM".

Reply: We fixed it.

Figure 4: subfigure keys (Ni, Al, Co...) should be placed in fix position in the top left corners.

Reply: We fixed it. Thank you for your help to make this paper approaching perfect.

Reviewer 3 Report

In this manuscript the authors studied phase transformations in a high entropy alloy during isothermal holding at elevated temperature. Major conclusions are properly backed by the examinations and the analysis techniques. However, introduction could be improved by revealing the importance of the study from practical point of view. Reasoning of material selection for the study, the practical use of the selected material and the anticipated changes in the properties of HEAs after phase transformation is also suggested. The stoichiometry or composition of the phases formed after transition should be also addressed. Why did those phases formed with the given composition? On what ground can this phase can be termed as austenite?

Author Response

In this manuscript the authors studied phase transformations in a high entropy alloy during isothermal holding at elevated temperature. Major conclusions are properly backed by the examinations and the analysis techniques. However, introduction could be improved by revealing the importance of the study from practical point of view. Reasoning of material selection for the study, the practical use of the selected material and the anticipated changes in the properties of HEAs after phase transformation is also suggested. The stoichiometry or composition of the phases formed after transition should be also addressed. Why did those phases form with the given composition? On what ground can this phase can be termed as austenite? 

Reply: It is very difficult to invent a new commercial alloy to replace the known commercial alloys. It is a very long way to go, waste of time and money. We just provided the basis of the knowledge to the HEAs which might have some commercial alloys to emerge in the near future. But who knows. I have just concentrated myself on the phase transformations of the metals and alloys, and fortunately found some interesting results of phase transformations in the HEAs. Nothing is related to the commercialized issues.

We replaced the austenite with FCC or FCC phase.

Reviewer 4 Report

1.line 65. It is desirable to describe the technology in more details. What were the “reannealings”, what was the number of cycles, what was the degree of deformation during forging and during rolling.

2. Fig.1c. It is desirable to comment in the text the presence of an additional peak at the angle of 11ï‚°.

3. Line 110. The text like “...along the 011 direction...” should be changed to “...along the <011> direction...” or “...along the [011] direction..”.

4. Line 111. Like the previous, “...the appearance of extra superlattice 100 reflections.” The symbol (100) should be used.

5. It seems to be useful to estimate and to describe briefly in the text shapes and sizes for Fig.3 particles.

Author Response

1.line 65. It is desirable to describe the technology in more details. What were the “reannealings”, what was the number of cycles, what was the degree of deformation during forging and during rolling.

2. Fig.1c. It is desirable to comment in the text the presence of an additional peak at the angle of 11ï‚°.

3. Line 110. The text like “...along the 011 direction...” should be changed to “...along the <011> direction...” or “...along the [011] direction.”.

4. Line 111. Like the previous, “...the appearance of extra superlattice 100 reflections.” The symbol (100) should be used.

5. It seems to be useful to estimate and to describe briefly in the text shapes and sizes for Fig.3 particles.

Reply:

1. We hot-forged the HEA at 1473 K. However, it cooled down very rapidly while being forged. Thus, we put it back to the furnace, re-annealed it to 1473 K, and forged it again. We did this cycle for at least three times to make sure the uniformity of the composition of the HEA. We did the cold rolling procedure at room temperature.

2. It is from the signal of the quartz tube which contained the HEA fine grains from grinding the HEA plate manually.

3. It is fixed: along the [011] direction.

4. It is fixed: (100) reflection.

5. We would like to estimate the particle shape as sphere, and the spherical particles are with a diameter of about 2 nm.

Round 2

Reviewer 3 Report

I think the manuscript can be improved by reasoning of the significance of revealing such phase transformation in HAEs and addressing its anticipated impact on the properties. Nevertheless, the several modifications made already as compared to the original version makes this manuscript acceptable for publication.